# Immunoinformatic Execution and Design of an Anti-Epstein–Barr Virus Vaccine with Multiple Epitopes Triggering Innate and Adaptive Immune Responses

**DOI:** 10.3390/microorganisms11102448

**Published:** 2023-09-29

**Authors:** Naveed Ahmed, Ali A. Rabaan, Ameen S. S. Alwashmi, Hawra Albayat, Mutaib M. Mashraqi, Ahmad A. Alshehri, Mohammed Garout, Wesam A. Abduljabbar, Nik Yusnoraini Yusof, Chan Yean Yean

**Affiliations:** 1Department of Medical Microbiology and Parasitology, School of Medical Sciences, Universiti Sains Malaysia, Kubang Kerian 16150, Malaysia; 2Molecular Diagnostic Laboratory, Johns Hopkins Aramco Healthcare, Dhahran 31311, Saudi Arabia; 3College of Medicine, Alfaisal University, Riyadh 11533, Saudi Arabia; 4Department of Public Health and Nutrition, The University of Haripur, Haripur 22610, Pakistan; 5Department of Medical Laboratories, College of Applied Medical Sciences, Qassim University, Buraydah 51452, Saudi Arabia; 6Infectious Disease Department, King Saud Medical City, Riyadh 12746, Saudi Arabia; 7Department of Clinical Laboratory Sciences, College of Applied Medical Sciences, Najran University, Najran 61441, Saudi Arabia; 8Department of Community Medicine and Health Care for Pilgrims, Faculty of Medicine, Umm Al-Qura University, Makkah 21955, Saudi Arabia; magarout@uqu.edu.sa; 9Department of Medical Laboratory Sciences, Fakeeh College for Medical Sciences, Jeddah 21461, Saudi Arabia; 10Institute for Research in Molecular Medicine (INFORMM), Universiti Sains Malaysia, Health Campus, Kubang Kerian 16150, Malaysia; 11Hospital Universiti Sains Malaysia, Universiti Sains Malaysia, Health Campus, Kubang Kerian 16150, Malaysia

**Keywords:** computational tools, vaccine designs, in silico, immunoinformatics, dry lab

## Abstract

One of the most important breakthroughs in healthcare is the development of vaccines. The life cycle and its gene expression in the numerous virus-associated disorders must be considered when choosing the target vaccine antigen for Epstein–Barr virus (EBV). The vaccine candidate used in the current study will also be effective against all other herpesvirus strains, based on the conservancy study, which verified that the protein is present in all herpesviruses. From the screening, two B-cell epitopes, four MHC-I, and five MHC-II restricted epitopes were chosen for further study. The refined epitopes indicated 70.59% coverage of the population in Malaysia and 93.98% worldwide. After removing the one toxin (PADRE) from the original vaccine design, it was projected that the new vaccine would not be similar to the human host and would instead be antigenic, immunogenic, non-allergenic, and non-toxic. The vaccine construct was stable, thermostable, soluble, and hydrophilic. The immunological simulation projected that the vaccine candidate would be subject to a long-lasting active adaptive response and a short-lived active innate response. With IgM concentrations of up to 450 cells per mm^3^ and active B-cell concentrations of up to 400 cells per mm^3^, the B-cells remain active for a considerable time. The construct also discovered other conformational epitopes, improving its ability to stimulate an immune response. This suggests that, upon injection, the epitope will target the B-cell surface receptors and elicit a potent immune response. Furthermore, the discotope analysis confirmed that our conformational B-cell epitope was not displaced during the design. Lastly, the docking complex was stable and exhibited little deformability under heat pressure. These computational results are very encouraging for future testing of our proposed vaccine, which may potentially help in the management and prevention of EBV infections worldwide.

## 1. Introduction

Epstein–bar virus (EBV) belongs to the herpesvirus family, which belongs to the most common human virus’s category. It was first discovered in 1964 in human cancer cells [1]. It is a double-strand DNA virus such as other herpes viruses. It has a large genome, which is almost 176 kb in length, which can translate to 80 different proteins, and it also has 46 untranslated RNAs [2]. Everyone worldwide becomes infected with this virus at least once in their lifetime. Its ability to persist for a lifetime in a person makes it a unique viral pathogen [3]. The EBV is transmitted through saliva, and primary infection usually occurs during childhood, which is mostly asymptomatic [4]. During adolescence, EBV can cause infectious mononucleosis [5]. It is a characteristic of all herpesviruses that they express their gene in a second mode, allowing them to persist in host cells for a long time without viral protein residue [6].

Primarily, EBV infects the B lymphocytes and sometimes epithelial cells. It was observed that if the primary infection did not happen at an early age, then the infection at a late age, known as infectious mononucleosis (IM), leads to excessive immune response with severe symptoms as a result [6]. It is still a dilemma why the immune response to IM is more severe than primary infection at an early age. Some theories of human leukocyte antigen (HLA) involvement are under consideration, such as that HLA may favor IM; it was also observed that if an initial viral dose is high, this could also be a reason for IM [7]. About 56 years ago, an association between cancer and EBV was discovered from a biopsy of Burkitt’s Lymphoma, which was endemic in Africa, and, apparently, it was the first virus that had a direct association with cancer in humans [8]. It is assumed that the long persistence of the virus occurs in B lymphocyte cells, and, in carriers, usually only 0.1% of cells are infected, but in the case of EBV-infected cancer cells, all infected malignant cells would lead to a strong association of EBV with cancer [9]. Cancer development may take years, and the long persistence of EBV may also explain the reason for cancer association. B lymphocyte cell differentiation triggered by EBV plays a role in developing EBV-related malignancies. Lymphoma, natural killer/T-cell lymphomas, and methotrexate-associated lymphomas are also among the malignancies associated with EBV.

Regarding understanding and analyzing immunological data, bioinformatics has developed a specialized subject known as computational immunology, sometimes known as immunoinformatics [10]. One of the most studied areas of applied immunology is using databases and other technologies to predict B- and T-cell epitopes. With advances in sequencing methods [11,12], researchers may now use an organism’s genomic information to discover vaccine candidates computationally, going beyond the traditional vaccinology approach. Vaccines have mostly focused on elements involved in infection establishment and host adverse effects, such as significant colonization factors, adhesion proteins, and other well-characterized virulence components. However, the pathogen’s genome encodes several as-yet-uncharacterized proteins that may encode antigenic areas. Immunoinformatics methods may be useful, particularly for illnesses for which little is known about the processes of pathogenesis or the antigenic determinants that cause them [13,14]. The methods used here were chosen to test the epitope peptides of outer glycoprotein and determine how well they might stimulate humoral and cellular immune responses.

## 2. Materials and Methods

### 2.1. Conservancy Analysis of the Target Protein

The major outer glycoprotein (886 amino acids), known as gp350/220 or the membrane antigen, of the herpesvirus was retrieved from NCBI (https://www.ncbi.nlm.nih.gov/protein/P68343/?report=fasta) (accessed on 6 August 2022) under the accession ID sp|P68343.1 for vaccine design, as it initiates the viral entry into the host cells. To ensure that the vaccine candidate derived from this protein will be efficacious to all the herpesvirus strains, we conducted a conservancy analysis using the Conservancy Domain Database of the NCBI (https://www.ncbi.nlm.nih.gov/Structure/cdd/cdd.shtml) (accessed on 6 August 2022). The database analyzed the conserved domains of the gp350/220 protein and whether it is present in all viral strains.

### 2.2. Characterization of Potential Epitopes

To determine the potential B-lymphocytes-, helper T-lymphocytes-(HTL), and cytotoxic T-lymphocytes-(CTL) specific epitopes, the linear B-cell epitope prediction window (http://tools.iedb.org/bcell) (accessed on 7 August 2022), structure-based conformational epitope prediction by Ellipro (http://tools.iedb.org/tools/ElliPro/iedb_input) (accessed on 7 August 2022) and Discotope (http://tools.iedb.org/discotope/) (accessed on 7 August 2022), MHC-I-restricted epitope prediction window (http://tools.iedb.org/mhci/) (accessed on 7 August 2022), and MHC-II-restricted epitope prediction window (http://tools.iedb.org/mhcii/) (accessed on 7 August 2022) were utilized.

#### 2.2.1. B-Lymphocytes-Specific Epitopes

For the linear epitopes, website accessed on 7 August 2022, different algorithms, including the Emini scale for surface accessibility [15], Karplus and Schulz scale for flexibility [16], Chou and Fasman algorithm for beta-turn [17], and Parker scale for hydrophilicity [18], are used by the IEDB. Ellipro predicts the conformational epitopes using the input structure’s protrusion index (PI) [19]. Epitopes with scores closer to 1 are the most outside the ellipsoid, making them easily accessible to the solvents and readily soluble. Discotope confirms the discontinuous epitopes predicted by Ellipro using surface measures and spatial neighborhood definition [20]. The protein structure was modeled using ITASSER (https://zhanggroup.org/I-TASSER/) (accessed on 7 August 2022) by providing the NCBI-retrieved protein sequences.

#### 2.2.2. CTL-Specific Epitopes

For MHC-I, the ANN 4.0 algorithm [21] was used on 8 August 2022 based on its capacity to predict and sort epitopes according to their IC_50_ values (in ascending order). All the alleles in the reference set were chosen to obtain as many restrictions as possible.

#### 2.2.3. HTL-Specific Epitopes

The NN-align 2.3 algorithm [21] was used for MHC-II as MHC class II interacting epitope cores, and affinity is determined simultaneously using this method. The NN-align development employs a unique training method for rectifying partiality in the training set brought on by duplicate binding core modeling [21]. The predictive performance is demonstrated to be much enhanced by including data about the residues around the peptide-binding region. All human HLA-DP, HLA-DQ, and HLA-DR alleles were used in the prediction with the default epitope length of 15. The software (version 2.3) was accessed on 9 August 2022.

### 2.3. Immunogenicity of the Selected Epitopes

All the shortlisted epitopes were further characterized based on their immunogenic potential using Vaxijen 2.0 (http://www.ddg-pharmfac.net/vaxijen/VaxiJen/VaxiJen.html) (accessed on 10 August 2022) for antigenicity, ToxinPred (https://webs.iiitd.edu.in/raghava/toxinpred/protein.php) (accessed on 10 August 2022), and AllerTOP (https://www.ddg-pharmfac.net/AllerTOP/index.html) for allergenicity accessed on 11 August 2022. The epitopes with a positive analysis for antigenicity and a negative analysis for allergenicity were finalized for the construct.

### 2.4. Evaluation of the Population Coverage of the Finalized Epitopes

The population coverage tool (IEDB; http://tools.iedb.org/population/) was accessed on 12 August 2022 to evaluate the general and local (Malaysian) population inclusivity. This software determines the percentage of people likely to react to a certain collection of epitopes based on their HLA genotype, MHC interaction, and T-cell limitation data [22]. The individual MHC-I and MHC-II epitope coverages and combined coverage were predicted for the world and the Malay populations.

### 2.5. Final Vaccine Construct

The finalized epitopes were aligned with the help of linkers (to maintain the individual epitope domains), an adjuvant (to improve the entry and immunogenicity of the vaccine candidate), and an additional peptide for cell penetration of the construct. Previous studies by Sanchez et al. (2021) [23] and Naveed et al. (2022) [24] were followed to design the construct.

### 2.6. Validation of the Immunogenicity of the Construct

The vaccine construct was subjected to antigenicity (using Vaxijen 2.0), allergenicity (using AllerTOP), druggability (using the PBIT server, http://www.pbit.bicnirrh.res.in/pipeline.php) (accessed on 13 August 2022), IFN-gamma stimulation (using the scan module of the IFNepitope server for MHC-II-restricted peptides, https://webs.iiitd.edu.in/raghava/ifnepitope/scan.php) (accessed on 13 August 2022), toxicity (through ToxinPred), host and host-gut microbe non-homology analysis (using the PBIT server), and proteasome cleavage analyses for MHC-I-restricted epitopes to evaluate whether, after cleavage in the host system, the same epitopes as predicted are obtained (using the processing prediction module of the IEDB server, http://tools.iedb.org/processing/) (accessed on 14 August 2022) to confirm its immunogenic, non-homologous, and non-toxic potency.

### 2.7. Modeling the Construct

The vaccine candidate was subjected to ITASSER on 15 August 2022 for tertiary structure prediction, and the model with a C-score closest to +2 was selected for further analysis. It was then refined using the GalaxyRefine of the GalaxyWEB server (https://galaxy.seoklab.org/) (accessed on 16 August 2022), and the model with the most improved RAMA score, least RMSD score, and most improved MolProbity score was finalized.

### 2.8. Physicochemical Properties and Structural Validation of the Construct

The construct was subjected to PSIPRED (http://bioinf.cs.ucl.ac.uk/psipred/), (accessed on 17 August 2022), to predict the secondary structure, nature of amino acids in the structure, gene ontology, protein disorder, and localization of the vaccine construct. Furthermore, ProteinSol and Scratch Protein Predictor’s SOLpro and DISpro (https://scratch.proteomics.ics.uci.edu/) were run on 17 August 2022 to predict the vaccine solubility and disordered regions, respectively. The Expasy ProtParam server (https://web.expasy.org/protparam/) was accessed on 18 August to predict its physicochemical properties, significantly the instability index validating the construct’s stability, the GRAVY index validating its hydrophilic/hydrophobic nature, and the half-life validating the time it takes to be eliminated from the host system. It was then validated for predicted secondary structure using the Ramachandran plot from different sources, including the Z-lab (https://zlab.umassmed.edu/bu/rama/) (accessed on 18 August 2022), the WHAT IF server (https://swift.cmbi.umcn.nl/servers/html/index.html) (accessed on 18 August 2022), and the MolProbity server of Duke University (http://molprobity.biochem.duke.edu/) for cross-validation.

### 2.9. Prediction of Immune Stimulation

To evaluate the predicted immune response against the designed vaccine candidate, the C-IMMSIM server was accessed on 19 August 2022 through https://kraken.iac.rm.cnr.it/C-IMMSIM/index.php?page=0. The construct sequence was provided in FASTA format, and no other changes were made.

### 2.10. Prediction of the Discontinuous Epitopes of the Final Construct

The IEDB server was utilized on 19 August 2022 to predict the discontinuous, conformational epitopes of the vaccine construct to ensure the surface accessibility of the vaccine candidate.

### 2.11. Binding Pocket and Molecular Interaction Analysis of the Construct

The TRAPP webserver (https://trapp.h-its.org/) was accessed on 20 August 2022 to predict the binding pockets of the vaccine candidate and the residues involved in potential molecular interactions. This assessment was further used to validate the molecular docking and interactions of the vaccine candidate with the Toll-like Receptor-2. The 3D structure of the TLR was taken from Uniprot (https://www.uniprot.org/) with protein code: O60603 accessed on 20 August 2022.

TLR-2 was selected because of its potential to interact with various exogenous pathogenic derivates, including LPS, lipopeptides, and peptidoglycan of bacterial pathogens; taxol, zymosan, mannan, etc. of fungal pathogens; glycoinositolphospholipids and hemozoin of parasites; and DNA, RNA, envelop proteins, and hemagglutinin proteins of viral pathogens [25]. Cluspro (https://cluspro.bu.edu/login.php?redir=/queue.php) was run on 21 August 2022 for the docking analysis. Cluspro identifies all possible molecular interactions and poses of the docked complex. The complex with the least energy (in negative value) was taken for the analysis of molecular interactions using PyMol (https://pymol.org/2/) accessed on 22 August 2022.

### 2.12. Molecular Dynamics Simulation

To further validate the docked complex’s thermostability, flexibility, and deformability potential, IMODs server (https://imods.iqfr.csic.es/) was utilized on 23 August 2022.

### 2.13. Codon Optimization and Expression Analysis

Since the vaccine candidate was designed for a human host, codon optimization was obtained by subjecting the vaccine candidate amino acid sequence to JCAT (http://www.jcat.de/) accessed on 24 August 2022. The pasted sequence was specified as ‘protein’, and ‘*Homo sapiens*’ were selected as the target organism. The raw and the optimized constructs were then checked for conservation using ClustalW (https://www.genome.jp/tools/clustalw/) accessed on 24 August 2022 to ensure that no amino acid had been changed during optimization. The EMBOSS transeq tool (https://www.ebi.ac.uk/Tools/st/emboss_backtranseq/) (accessed on 25 August 2022) was used to reverse translate the protein sequence for expression analysis. pET 28a (+) was used from the Snapgene plasmid resource as a plasmid for the expression, and the analysis was performed using the Snapgene offline software, downloaded from https://www.snapgene.com/ accessed on 25 August 2022.

## 3. Results

### 3.1. Conservancy Analysis of the Target Protein

The conservancy analysis confirmed that the protein is confirmed in all herpesviruses, thus a vaccine candidate against the EBV will also be effective against all the other herpesvirus strains. The analysis further demonstrated that the protein with ID pfam5109 is the only member of the cl37540 superfamily and spans multiple domains, all conserved throughout the herpesvirus strains. The protein was antigenic (0.5695, at a threshold of 0.5), non-allergenic, and non-toxic.

### 3.2. Characterization of Potential Epitopes

Two B-cell epitopes, four MHC-I-restricted epitopes, and five MHC-II-restricted epitopes were finalized from the screening. The findings are summarized in Table 1. The screening and finalization process is as follows:

#### 3.2.1. B-Lymphocytes-Specific Epitopes

The B-lymphocyte-specific epitopes were finalized based on their immunogenic potential in continuous and discontinuous conformations. The first epitope was a constant linear epitope and the second was discontinuous. However, both were evaluated as conformational. Both the epitopes were predicted as surface accessible, flexible, more beta-turns, hydrophilic, antigenic, non-allergenic, and non-toxic. Figure 1 provides the graphical representations of the modules run for the epitopes. The residues in yellow illustrate antigenic and immunogenic character, while the green color depicts non-immunogenic residues (Appendix A).

#### 3.2.2. CTL-Specific Epitopes

Out of 27,532 allele restrictions, 7 epitopes were shortlisted based on IC_50_ values below 50 and ranked between 0 and 10. These were further screened according to antigenicity, allergenicity, individual Malay and world coverage, immunogenicity, and toxicity analyses, providing four final epitopes. The finalized epitopes and their restricting MHC-I alleles are provided in Table 1.

#### 3.2.3. HTL-Specific Epitopes

Out of 48,923 allele restrictions, 13 epitopes were shortlisted based on IC_50_ values below 50 and ranked between 0 and 10. These were further screened according to antigenicity, allergenicity, individual Malay and world coverage, immunogenicity, and toxicity analyses, providing five final epitopes. The finalized epitopes and their restricting MHC-II alleles are provided in Table 1.

### 3.3. Evaluation of the Population Coverage of the Finalized Epitopes

The finalized epitopes, subjected to the population coverage tool of the IEDB server, showed 70.59% Malaysian population coverage and 93.98% world population coverage, as shown in Figure 2A and Figure 2B, respectively. The analysis confirmed that the vaccine candidate would be productive in all parts of the world.

### 3.4. Final Vaccine Construct

The final vaccine construct shown in Figure 2C utilized the TAT peptide, the 50S ribosomal protein, the pan HLA DR-binding epitope, and linkers to fuse all the epitopes while maintaining their domains (B-cell, HTL, and CTL). The transactivator of transcription (TAT) peptide (GRKKRRQRRRPQ) is a cell-penetrating peptide isolated from the human immunodeficiency virus. Cell-penetrating peptides (CPPs) such as the TAT peptide transport large molecules across the lipophilic boundary of cell membranes and into the cell, where they can exert their biological effects [23]. HIV TAT peptide, which penetrates cells, is located at the N-terminal end of the multi-epitope construct. Then, the 50S ribosomal protein L7/L12 was selected to boost immunogenicity (Accession no. P9WHE3) following the study of Naveed et al. (2022) [10].

Pan DR epitope (PADRE-AKFVAAWTLKAAA) adjuvant was initially fused to serve as an HTL stimulus. Methionine was used at the start of the construct, linker EAAAK was used to fuse methionine with the TAT peptide, and linker GGGS was used to fuse the TAT peptide with the 50S ribosomal protein. PADRE was removed in the final construct since it was predicted to be toxic, according to ToxinPred. The 50S ribosomal protein was directly fused with the CTL epitopes (as an adjuvant and an HTL stimulator) using the linker GGGS, and GPGPG was used in between the epitopes. The HEYGAEALERAG linker was used in between the CTL, B-cell, and HTL epitopes, whereas GPGPG served as a linker in the inter-B-cell epitopes and AAY served to fuse inter-HTL epitopes. The HEYGAEALERAG was finally used to fuse the last epitope with the hexahistidine tag.

### 3.5. Validation of the Immunogenicity of the Construct

The initial vaccine construct had one toxin component, PADRE, which was then removed, and the final vaccine construct was predicted to be antigenic, immunogenic, non-allergenic, non-toxic, and non-homologous to the human host. It was also predicted to be druggable (Appendix A) through its significant similarity to the other commercially available drugs. The host non-homology analysis provided that the construct was non-homologous to the gut microbiota of the human host and the host itself, confirming that the immune response will not be undermined due to host tolerance to the vaccine candidate. IFN-gamma analysis predicted that four out of five MHC-II-restricted epitopes can stimulate IFN-gamma cytokines. Lastly, the proteasome analysis also validated the MHC-I-restricted epitopes with minimal IC_50_ values. Table 2 provides the values for individual immunogenic analyses.

### 3.6. Modeling the Construct

ITASSER provided five models of the vaccine candidate’s tertiary structures based on similarity with the tertiary structures present in the database. The first model with a C-score (confidence score) of 0.78 was considered. A model with −2 to +2 C-score on ITASSER is deemed fit to use. However, if the score is below +1, further refinement is needed. The vaccine candidate structure was subjected to GalaxyRefine, which further provided five models based on improved RAMA scores, molprobity scores, decreased RMSD values, and fewer poor rotameters. Model 2 was finalized and downloaded based on the best average of all the above-discussed scores. The RAMA score for the original (raw) vaccine candidate’s tertiary structure was 61.2%, whereas the refined model’s RAMA value was 83.9%. The raw and refined models are shown in Figure 3A, superimposed on each other to visualize the structural changes.

### 3.7. Physicochemical Properties and Structural Validation of the Construct

The analysis predicted that the vaccine construct was stable, thermostable, soluble, and hydrophilic, as discussed in Table 2. The solubility of the vaccine candidate was different when computed in different servers, but all predicted it to be soluble, as seen in Figure 3B. The localization analyses illustrated that it is a transmembrane protein, indicating its potential to be immunogenic inside the cell and its capacity to signal cytoplasmic immune signals (Figure 3C and Figure 4A). Figure 3E provided the cartoon structure of the vaccine candidate, whereas the disorder plot provided in Figure 3D illustrated that the amino acids with values greater than the cutoff point (0.5) were disordered. However, the construct was considered stable, as the majority of the amino acids were ordered. Ramachandran plots were predicted using different servers (Appendix A) to cross-validate the structure, and it was shown that most of the amino acids (88.281%) were in the highly preferred regions, 7.812% were plotted in the preferred regions, and only 3.906% were plotted in the questionable regions; the amino acids plotted in the questionable regions were the disordered residues shown in Figure 3E. Table 3 provides the gene ontology functions of the vaccine candidate for its potential use in other processes.

### 3.8. Prediction of Immune Stimulation

The immune simulation predicted a short-living active innate and a long-lasting active adaptive response against the vaccine candidate. The B-cell population graph (Figure 5A) showed that the B-cells stay active for a long time with the IgM concentrations of up to 450 cells per mm^3^ and the active B-cell concentrations of up to 400 cells per mm^3^. Figure 5B, showing the B-cell population (internalized vs. presented), illustrated a stable active response of up to 450 cells per mm^3^, with little to no anergic cell production. The presenting B-cell escalated to 350 cells per mm^3^ in the first few days of injection but declined rapidly as our antigen was eliminated within the first few days. The internalized B-cell population showed a similar response in Figure 5B and increased to 75 cells per mm^3^ before declining quickly along with the antigen and the presenting cells.

The helper T-cell graph shown in Figure 5C illustrated the TH memory cell count of up to 375 cells per mm^3^, gradually falling to around 275 cells per mm^3^ on the 350th day of injection, validating memory development for years. The non-memory helper T-cells increased to 4700 cells per mm^3^ in the first 25 days of injection and rapidly decreased afterward. The cytotoxic T-cells demonstrated a turbulent response (Figure 5D), wherein the non-memory population initially increased to 1160 cells per mm^3^ and then fluctuated between 1055 cells per mm^3^ and 1140 cells per mm^3^ for almost a year. No memory TC cells were recorded. As discussed earlier, the innate immune response was short-lived, with the NK-cell population fluctuating between 390 cells per mm^3^ and 310 cells per mm^3^ throughout the year (Appendix A). The active macrophages increased up to 100 cells per mm^3^ in the first 40 days of injection, then instantly fell to 25–30 cells per mm^3^ for the rest of the year (Appendix A). The resting macrophages recorded were around 150–200 cells per mm^3^ throughout, indicating that an adequate innate immune response will be generated even after the antigen enters the host system years after vaccine injection.

The DC population shown in Appendix A elucidated that the active, presenting, and internalizing dendritic cells were activated in the first few days of injection, with the count of active DC cells fluctuating between 20 and 30 cells per mm^3^ for the rest of the year, while the presenting and internalizing DCs disappearing after 25 and 5 days of injection, respectively. Appendix A showed the elimination of the antigen within the first 24 h of injection, validating the half-life (4.4 h) analysis in Table 2. It further depicted that the immune complex of IgM+IgG increased up to 10,000 (arbitrary value) as soon as the vaccine candidate was injected but rapidly decreased to 1000 for the rest of the year. IgM levels escalated to 6000 initially, and the levels of immune complex IgG1+IgG2 increased up to 4500, both decreasing steadily and disappearing after 140 days of injection. The body produced a minute danger signal upon the vaccine candidate injection, shown in Appendix A, which was cleared as soon as the antigen was eliminated. The high IFN-gamma value (425,000 mg/mL) supported our IFN-gamma stimulation results shown in Figure 2 and Table 2 and depicted a strong prompt of cytokines, as seen in Appendix A.

### 3.9. Prediction of the Discontinuous Epitopes of the Final Construct

The discotope analysis confirmed that our conformational B-cell epitope was not displaced during the design (shown in Figure 6A). However, other conformational epitopes were also found in the construct, improving its immune response stimulation potential, indicating that upon injection, the epitope will target the B-cell surface receptors and elicit a strong immune response.

### 3.10. Binding Pocket and Molecular Interaction Analysis of the Construct

The TRAPP server predicted 20 binding site pockets of the vaccine construct discussed in Table 4. Cluspro predicted 29 models of the TLR-2 (Figure 6B) and vaccine candidate models’ interactions, the 5th model was selected (provided in Figure 6C) based on the lowest energy (−1139.28 kcal/mol), and the interactions (provided in Table 5) visualized on PyMOL matched the binding pockets prediction of the TRAPP server. A total of 333 interactions were predicted using PyMOL, of which 17 were conventional bonds, 7 were hydrogen bonds, and 8 were transition bonds, whereas 1 was a salt-bridge, and 11 interactions are shown in Figure 6D–L.

### 3.11. Molecular Dynamics Simulation

The docking complex was stable and showed minimum deformability under thermal pressure, as shown in Appendix A. It is clearly indicated that the vaccine candidate was slightly bent under stereochemical and thermal pressure but did not have any significant effect on the structure overall. The deformability graph shown in Appendix A indicated the deformable and flexible loci of the complex. The B-factor graph (Appendix A) illustrated the mobile and static residues of the complex along with atomic displacements in an equilibrium conformation (Appendix A). The peaks show the mobile residues and are consistent with the deformability graph (Appendix A), indicating the flexibility of these residues. Since the structure is not recorded in the PDB database, no comparative plot was generated. The eigenvalue required to fluctuate the mode (through an inverse relationship) was 1.891723 × 10^−5^, and a higher eigenvalue indicated that the complex is stable and requires a great amount of energy to fluctuate or deform, shown in Appendix A. The variance (%), covariance, and elastic network map were depicted in Appendix A, respectively, elucidating the co-related residues (red), anti-corelated residues (white), and non-corelated residues (blue) in the covariance map, whereas the elastic network map showed the linked atoms of the complex, darker grey represented strong linkage.

### 3.12. Codon Optimization and Expression Analysis

A CAI value of 1.000 and GC content of 55.6995% were recorded upon codon optimization (Appendix A) of the construct for expression in the human host. The CLUSTALW analysis showed 100% similarity in pre- and post-optimization protein sequences (Appendix A). The cloning of the vaccine candidate in the pET28 (+) vector after reverse translating the protein sequence is indicated in Figure 7, with the linear map depicted in Appendix A. *Sal*I and *Bam*H1 restriction sites were used for the cloning, and overhangs were removed to convert the sticky ends to blunt ends for stable cloning. The history of in silico cloning has been shown in Appendix A.

## 4. Discussion

The most common infectious diseases with pandemic potential are viruses. They occasionally result in outbreaks in human populations linked to various animal reservoirs. Around 95% of people on the planet are carriers of the EBV, and most have no idea they have it. However, various factors might contribute to the creation of microenvironments linked to the development of certain illnesses [6]. The EBV-infected people who have shown symptoms, usually adults, recover within 2 to 4 weeks. On the other side, some individuals may experience fatigue for weeks or even months. Following an EBV infection, the virus remains dormant (inactive) in the host’s body. The virus may reappear under certain conditions [7]. Reactivation of EBV does not always result in symptoms. However, immunocompromised individuals are more prone to suffer symptoms [8]. The vaccination practices remain a potent preventive strategy to prevent any infections. Hence, in the current study, we have designed a vaccine candidate for immunization against EBV infections worldwide.

Chemotherapy and radiation, effective therapeutic methods for oncogenic cases related to EBV, is a gold standard method [26]. However, 15–30% of NPC patients have a gloomy outlook and a history of disaster in several places, while 5–15% have failed locally. In addition, radiotherapy and chemotherapy have many side effects [27]. As a result, finding a new therapeutic drug with few side effects and low toxicity of the target is a prominent topic worldwide. Cancer immunotherapy based on new vaccines appears as a realistic and successful treatment choice for various ferocity. The purpose of producing a vaccine related to EBV infections can be imagined because of the specific viral immunology and interaction with cancer cells [28]. Specific proteins related to EBV must be considered as a potential target for developing vaccines and immunological regulations [29]. A therapeutic vaccine has been studied in preclinical and clinical research, with promising results despite several challenges [29]. We concentrated on one of the outer glycoproteins in our study since it is essential for the structural and pathogenic functions of the virus. The protein might create an effective vaccine, according to analysis of its physical and chemical characteristics.

Immunization efforts in NPC focus on EBV proteins LMP1, LMP2, and EBNA1 [30]. Due to high expression levels, LMP2A and EBNA1 are the most attractive candidates for developing EBV-particular vaccines [31]. Because of maintaining virus DNA in proliferation cells and regulating biological processes, EBNA1 is the protein needed in NPC. It has a variety of CD4+T-cell epitopes, making it an immunotherapy target, unlike the others [32]. In the previous decade, several clinical studies on the satisfying efficiency of inoculation in NPC released to EBV revealed promising results. However, the current study showed that our target protein is also confirmed in all herpesviruses, thus a vaccine candidate against the EBV will also be effective against all the other herpesvirus strains. The protein was antigenic, non-allergenic, and non-toxic.

In healthy seropositive individuals, it was found that exposure to antigen-presenting cells (APCs) in vitro against fusion protein containing EBNA1 carboxyl terminals combined with LMP2 in poxvirus vectors produces efficacious renaissance of CD8+ LMP2 T-cells and EBNA-specific memory cells 1 [33]. A MVA-EBNA1/LMP2 therapeutic fusion vaccine is tested in two primary phase 1 clinical trials in NPC patients [31]. This inoculation is designed to mimic the immunostimulatory properties of EBNA1 and LMP2. The vaccine is a CD4+ and CD8+ synthesis protein containing an unfunctionally inactive epitope [34].

To check which epitope covers the Malay as well as the maximum world population, we used the population coverage analysis tool integrated in the IEDB server. The individual MHC-I and MHC-II epitope coverage and combined coverage were predicted for the world and the Malay population. This was because the distribution of MHC alleles varies between geographical or ethnic groups throughout the world. Because there are over one thousand different human MHC alleles, vaccination is only effective in those who have a certain MHC allele that binds the epitope. As a result, the tool of IEDB population coverage predicted that the vaccination would cover 70.59% of the Malaysian population and 93.98% of the world population. The analysis confirmed that the vaccine candidate would be productive in all parts of the world.

The evidence from emerging studies suggests that in addition to the ability of disease-specific effects, the vaccines have important non-specific effects (NSEs) [35]. These NSEs contribute to the overall effect on mortality and morbidity. The immunological studies have added plausibility to the existence of NSEs by showing that vaccines can train the innate immune system to increase resistance towards unrelated pathogens [36]. The results of the current study showed that the vaccine candidate has obtained effective immunogenic potentials, i.e., having antigenic, non-allergenic, and non-toxic properties, which can be used alone or in combination with other vaccine targets to construct a finely tuned multi-epitope vaccine for EBV. Furthermore, the immune simulation predicted a short-living active innate and a long-lasting active adaptive response against the vaccine candidate. The vaccine design in the current study had significant potential and could be evaluated for further wet lab experimental studies.

**Study limitations:** The current study was performed in silico, and no wet lab experiments were carried out to empower the reported outcomes.

## 5. Conclusions

To reduce the occurrence of infections associated with EBV, an effective therapeutic or preventive approach is required. As a result, an immunoinformatics-based vaccine was designed using chosen epitopes. By developing an epitope-based vaccination employing the outer glycoproteins as an immunogenic target by adopting an immunoinformatic method, the present study construct offers a clinical advance in response to EBV infections to limit its spread and pathogenesis. Immune modeling and examining physicochemical characteristics showed how effective the vaccine design was. The intended physicochemical and immunological responses are elicited by the vaccination design. It was demonstrated that the vaccine design led to an immune response aligned with our immunological simulation objectives. The construct may be identified by immune cell receptors, depending on the protein–protein interaction. The PET28a (+) plasmid was used to clone a construct, demonstrating the vaccine peptide’s good expression level. With this integrated computational method, experiments are guided more efficiently, with fewer repetitions and a lower risk of error. However, experimental validations (both in vitro and in vivo) can guarantee the effectiveness of any medication or vaccine. With various serological tests carried out to confirm the reaction trigger on demand, this framework should be considered as a possible candidate for in vitro and in vivo study in contraindications to EBV.

## Figures and Tables

**Figure 1 microorganisms-11-02448-f001:**
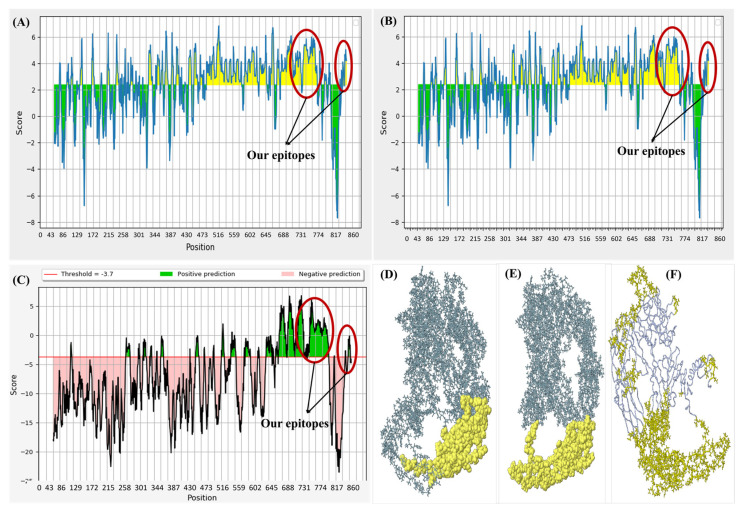
Graphical representation of the immunogenic epitopes. (**A**) Hydrophilicity; (**B**) linear epitope prediction. The green residues in all these graphs represent the non-epitope residues, while the ones in yellow represent the potential epitopes. (**C**) Discotope prediction of the conformational and discontinuous epitopes of the protein. The residues in green represent discontinuous but conformational epitopes; (**D**) Ellipro illustration of the first B-cell epitope, residues 699–733, as conformational and surface accessible; (**E**) Discotope 2D validating the Ellipro prediction; (**F**) 3D representation of all the conformational epitopes of the protein. The residues on the plots encircled in red represent the two finalized epitopes.

**Figure 2 microorganisms-11-02448-f002:**
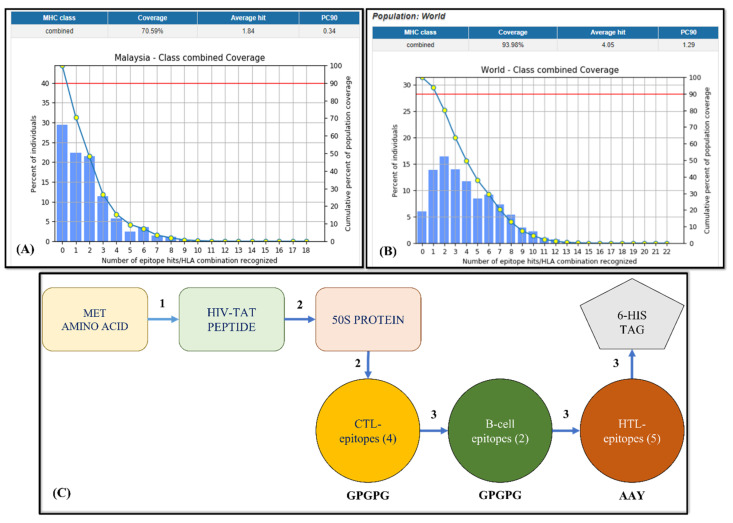
(**A**) Malaysian population coverage of the vaccine candidate; (**B**) world population coverage of the vaccine candidate; (**C**) the vaccine construct, where 1 represents EAAAK, 2 represents GSSS, and 3 represents HEYGAEALERAG linkers. Red line: minimum number of epitope hits / HLA combinations recognized by 90% of the population.

**Figure 3 microorganisms-11-02448-f003:**
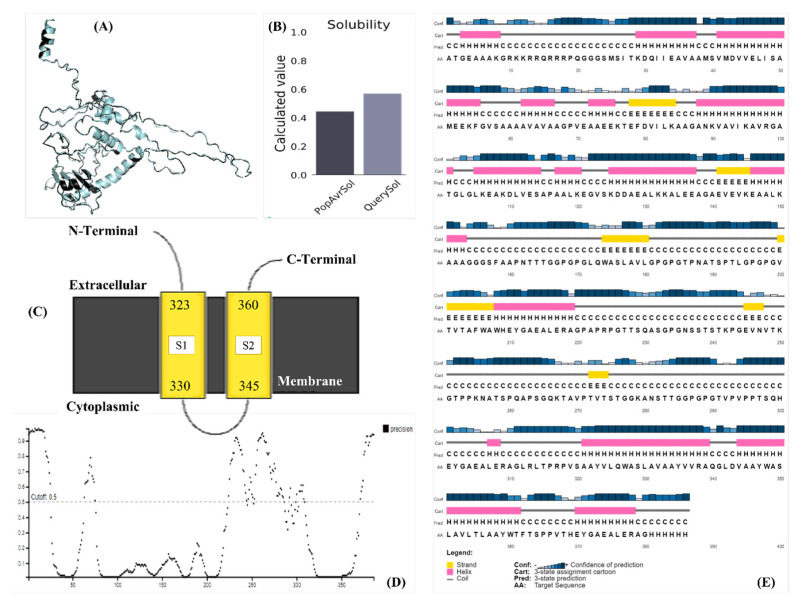
Modeling and physicochemical analysis of the construct; (**A**) the original and refined models superimposed on each other to visualize the minor structural difference; (**B**) protein solubility compared to the average solubility of proteins in *E. coli* using Protein sol; (**C**) localization of the vaccine construct; (**D**) disorder plot of the construct; (**E**) cartoon of the predicted secondary structure of the vaccine candidate.

**Figure 4 microorganisms-11-02448-f004:**
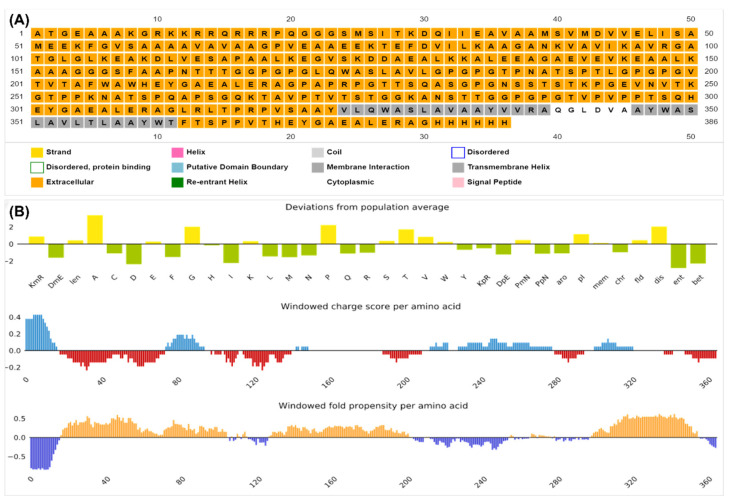
(**A**) MEMSTAT of the protein translating the localization shown in Figure 4; (**B**) vaccine candidate solubility predicted by PSIPRED.

**Figure 5 microorganisms-11-02448-f005:**
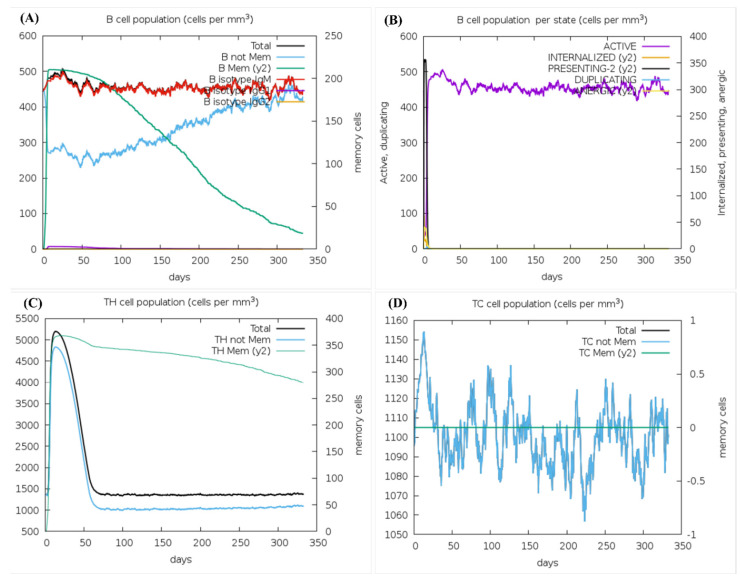
Immune simulation analysis. (**A**) B-cell population; (**B**) B-cell population per state; (**C**) TH (Helper-T) cell population; (**D**) TC (Cytotoxic-T) cell population.

**Figure 6 microorganisms-11-02448-f006:**
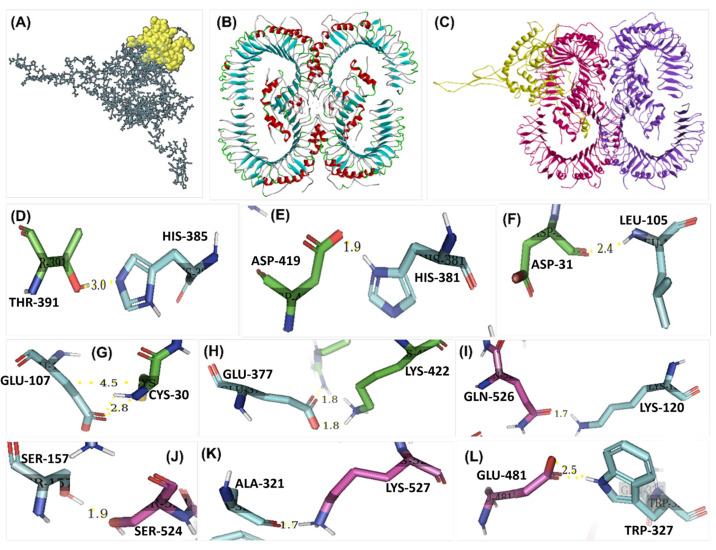
(**A**) The conformational B-cell epitope predicted earlier is conserved in the final vaccine construct; (**B**) the 3D structure of TLR-2 (protein code: O60603) obtained from UniProt KB; (**C**) the docked complex, where the purple-colored chains are the non-interacting chains and the ones in pink have been predicted to interact with the vaccine candidate, whereas the vaccine candidate is shown in yellow color; (**D**–**L**) some of the molecular interactions showing bond length and AA residue involved, where the blue residues belong to the vaccine candidate and the pink/green residues belong to TLR-2.

**Figure 7 microorganisms-11-02448-f007:**
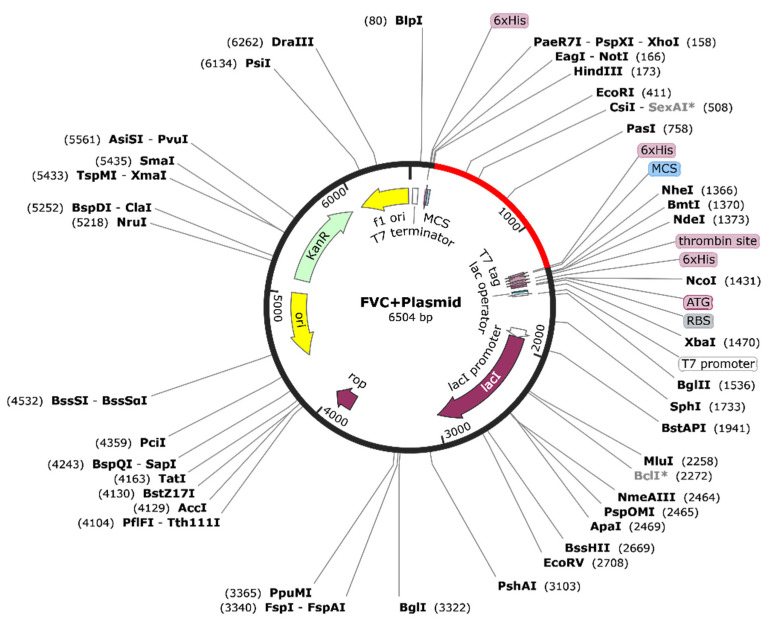
The pET28 (+) plasmid with the insert.

**Table 1 microorganisms-11-02448-t001:** Finalized B-cell, MHC-I, and MHC-II epitopes; the values in brackets illustrate the threshold value for each factor. All of the epitopes were non-allergen and non-toxic.

**Epitopes**	**Antigenicity (0.4)**	**B-Turn (1.061)**	**Hydrophilicity (2.382)**	**Flexibility (1.023)**	**Accessibility (1.000)**
B-lymphocytes-specific epitopes					
PAPRPGTTSQASGPGNSSTSTKPGEVNVTKGTPPKNATSPQAPSGQKTAVPTVTSTGGKANSTTG	0.984	1.702	4.7372	1.076	1.323
TVPVPPTSQ	1.1033	1.1633	2.873	1.225	1.633
**Epitopes**	**Antigenicity**	**Restricting HLA Alleles**			
MHC-I-restricted epitopes					
FAAPNTTTG	0.6798	HLA-B*35:01, HLA-C*03:03, HLA-C*12:03			
LQWASLAVL	1.6217	HLA-A*02:06, HLA-B*15:01, HLA-B*48:01			
TPNATSPTL	0.4187	HLA-B*39:01, HLA-B*35:01			
VTVTAFWAW	0.6731	HLA-B*58:01, HLA-B*57:01			
MHC-II-restricted epitopes					
LRLTPRPVS	2.6164	HLA-DQA1*02:01/DQB1*04:02, HLA-DPA1*01:03/DPB1*03:01, HLA-DQA1*05:01/DQB1*04:02, HLA-DRB1*11:01, HLA-DRB1*13:01, HLA-DRB3*03:01, HLA-DRB1*08:02, HLA-DRB3*02:02, HLA-DRB1*08:01, HLA-DRB1*03:01, HLA-DQA1*01:02/DQB1*05:01			
VLQWASLAV	0.9940	HLA-DQA1*02:01/DQB1*03:03, HLA-DPA1*03:01/DPB1*04:02, HLA-DRB4*01:01, HLA-DRB1*04:05, HLA-DRB1*15:01, HLA-DRB4*01:03, HLA-DQA1*01:02/DQB1*05:01, HLA-DRB1*13:01, HLA-DPA1*01:03/DPB1*06:01, HLA-DQA1*02:01/DQB1*03:01			
VVRAQGLDV	0.8213	HLA-DRB4*01:01, HLA-DRB1*09:01, HLA-DRB1*07:01, HLA-DRB1*13:02, HLA-DRB3*03:01, HLA-DRB1*01:01, HLA-DRB4*01:03			
WASLAVLTL	1.0200	HLA-DRB1*09:01, HLA-DRB1*10:01, HLA-DRB4*01:03, HLA-DRB1*13:01			
WIFTSPPVT	0.4742	HLA-DQA1*06:01/DQB1*04:02, HLA-DQA1*02:01/DQB1*04:02, HLA-DRB1*07:01, HLA-DQA1*05:01/DQB1*04:02, HLA-DRB1*10:01, HLA-DRB1*01:01, HLA-DQA1*01:02/DQB1*05:01			

**Table 2 microorganisms-11-02448-t002:** Individual immunogenic and physicochemical analyses of the vaccine construct and their interpretation.

Analysis	Result; Score (Threshold)
Antigenicity	Antigenic; 0.5710 (0.5000)
Allergenicity	Non-allergen
Toxicity	Non-toxic
IFN-gamma stimulation for epitope 1	Positive; 0.318 (0.000)
IFN-gamma stimulation for epitope 2	Positive; 0.161 (0.000)
IFN-gamma stimulation for epitope 3	Positive; 0.555 (0.000)
IFN-gamma stimulation for epitope 4	Positive; 0.162 (0.000)
IFN-gamma stimulation for epitope 5	Negative; −0.8088 (0.000)
Non-homology analysis against human proteome	Non-homologous
Non-homology analysis against gut microbiota	Non-homologous
No. of amino acids	386
Molecular weight	38,966.82
Theoretical pI	8.06
Estimated half-life in mammalian reticulocytes	4.4 h
Instability index	Stable; 33.94 (<34)
Aliphatic index	Thermostable; 73.76
GRAVY	Hydrophilic; −0.195 (<0)
Solubility upon overexpression (Scratch)	Soluble; 0.955 (0.5)

**Table 3 microorganisms-11-02448-t003:** Gene ontology functions of the vaccine construct predicted by PSIPRED.

GO Term	Name	Prob
Biological Process		
GO:0006396	RNA processing	0.6
GO:0010468	regulation of gene expression	0.6
GO:0000398	mRNA splicing, via spliceosome	0.62
GO:0009059	macromolecule biosynthetic process	0.63
GO:0006351	transcription, DNA-templated	0.65
GO:0006355	regulation of transcription, DNA-templated	0.7
GO:0051252	regulation of RNA metabolic process	0.71
GO:0006810	transport	0.71
GO:0008380	RNA splicing	0.73
GO:0051171	regulation of nitrogen metabolic process	0.73
GO:0034645	cellular macromolecule biosynthetic process	0.8
GO:2001141	regulation of RNA biosynthetic process	0.8
GO:0019222	regulation of metabolic process	0.8
GO:1903506	regulation of nucleic acid-templated transcription	0.82
Molecular Functions		
GO:0003676	nucleic acid binding	0.97
GO:0044822	poly(A) RNA binding	0.87
GO:0008092	cytoskeletal protein binding	0.82
GO:0003723	RNA binding	0.8
GO:0000166	nucleotide binding	0.79
GO:0019900	kinase binding	0.74
GO:0003824	catalytic activity	0.65
GO:0003779	actin binding	0.64
GO:0015631	tubulin binding	0.63
GO:0003677	DNA binding	0.63
GO:0008017	microtubule binding	0.6
GO:0032549	ribonucleoside binding	0.59
GO:0001664	G-protein coupled receptor binding	0.59
GO:0001883	purine nucleoside binding	0.58
GO:0017076	purine nucleotide binding	0.56
GO:0035639	purine ribonucleoside triphosphate binding	0.55
GO:0016817	hydrolase activity, acting on acid anhydrides	0.52
Cellular Functions		
GO:0005739	mitochondrion	0.83
GO:0031224	intrinsic component of membrane	0.71
GO:0016020	membrane	0.7
GO:0031966	mitochondrial membrane	0.61
GO:0005886	plasma membrane	0.58
GO:0016021	integral component of membrane	0.53
GO:0030529	ribonucleoprotein complex	0.53

**Table 4 microorganisms-11-02448-t004:** Binding pockets of the vaccine candidate for molecular interactions. A represents the chain of the molecule here.

Score	Coord_x	Coord_y	Coord_z	Residues
0.44332	101.61	93.8105	84.3158	A_43_ASP; A_47_LEU
0.3123	74.7601	89.5706	139.954	A_372_GLY; A_375_ALA; A_376_LEU; A_379_ALA; A_380_GLY
0.30649	98.8762	95.9583	81.1978	A_48_ILE; A_51_MET
0.27802	103.259	96.5484	87.0455	A_141_GLU; A_178_ALA; A_179_VAL; A_180_LEU
0.24589	69.878	91.1676	135.446	A_369_HIS; A_374_GLU; A_375_ALA; A_378_ARG; A_379_ALA
0.24362	75.9403	88.6986	105.536	A_98_ARG; A_102_GLY; A_103_LEU; A_104_GLY; A_105_LEU; A_314_LEU; A_315_THR; A_316_PRO; A_317_ARG
0.2169	97.332	93.4671	84.5491	A_51_MET; A_137_GLU; A_139_GLY; A_141_GLU
0.21223	101.958	100.489	85.6208	A_179_VAL; A_180_LEU; A_181_GLY; A_206_TRP
0.19845	105.508	92.7975	87.0885	A_8_LYS; A_43_ASP; A_170_PRO; A_174_TRP; A_178_ALA
0.15525	72.1554	84.3839	97.2013	A_105_LEU; A_310_ALA; A_311_GLY; A_312_LEU; A_313_ARG
0.14171	74.9699	89.9079	109.34	A_103_LEU; A_104_GLY; A_316_PRO; A_317_ARG
0.14057	104.133	97.4513	82.7488	A_43_ASP; A_180_LEU
0.13555	111.707	102.763	81.6146	A_36_VAL; A_208_TRP; A_230_GLN
0.1255	74.1947	82.6675	101.329	A_98_ARG; A_105_LEU; A_311_GLY; A_312_LEU; A_313_ARG
0.12307	69.1664	88.2258	141.794	A_378_ARG; A_379_ALA; A_380_GLY; A_381_HIS; A_382_HIS; A_383_HIS; A_384_HIS
0.12226	106.456	78.4865	93.9053	A_9_GLY; A_10_ARG; A_13_ARG; A_255_LYS
0.11209	91.481	91.5907	100.331	A_127_ASP
0.10931	109.269	99.044	83.7031	A_36_VAL; A_40_SER; A_178_ALA; A_180_LEU; A_208_TRP; A_210_GLU
0.1084	106.421	101.195	80.0099	A_180_LEU; A_208_TRP
0.10469	75.056	82.1306	110.437	A_349_ALA; A_350_SER; A_351_LEU

**Table 5 microorganisms-11-02448-t005:** Predicted molecular interactions between the vaccine candidate and TLR-2.

Interacting Residues				
Sr.	Vaccine AA Residue	Receptor AA Residue	Bond Length (Angstrom)	Bond Type
1	His385	Thr391	3.0	Amine (hydrogen)
2	His381	Asp419	1.9	Conventional covalent bond
3	Leu105	Asp31	2.4	H-bond–van der Waals transition
4	Glu107	Cys30	2.8	Hydrogen
5	Glu377	Lys422	1.8	Conventional covalent bond
6	Glu377	Lys422	1.8	Conventional covalent bond
7	Lys120	Glu526	1.7	Conventional covalent bond
8	Ser157	Ser524	1.9	Conventional covalent bond
9	Ala321	Lys527	1.7	Conventional covalent bond
10	Trp327	Glu481	2.5	H-bond–van der Waals transition
11	Trp327	Glu481	2.0	H-bond–van der Waals transition
12	Tyr371	Arg486	2.0	H-bond–van der Waals transition
13	Tyr371	Arg486	2.7	Hydroxyl (hydrogen)
14	Gln326	Asp419	2.0	H-bond–van der Waals transition
15	Gln326	Typ440	2.7	Hydroxyl (hydrogen)
16	Ser326	Arg340	1.9	Conventional covalent bond
17	Leu330	Arg340	2.4	H-bond–van der Waals transition
18	Lys242	Glu310	1.8	Conventional covalent bond
19	Lys242	Glu310	1.9	Conventional covalent bond
20	Pro221	Lys308	1.7	Conventional covalent bond
21	Arg10	Asn199	2.6	Carboxylic (hydrogen)
22	Arg10	Asn199	1.8	Conventional covalent bond
23	Glu173	Glu225	2.7	Hydroxyl (hydrogen)
24	Arg338	Glu152	2.5	H-bond–van der Waals transition
25	Arg338	Glu152	1.8	Conventional covalent bond
26	Ala345	Glu103	2.9	Oxazole (hydrogen)
27	Tyr335	Glu177	1.9	Conventional covalent bond
28	Arg338	Glu177	1.8	Conventional covalent bond
29	Tyr371	Arg508	1.9	Conventional covalent bond
30	Tyr371	Arg508	1.8	Conventional covalent bond
31	Tyr371	Arg508	1.8	Conventional covalent bond
32	Ala375	Lys505	2.1	H-bond–van der Waals transition
33	Glu107	Cys30	4.5	Salt bridge

## Data Availability

More data related to the current study would be available upon a reasonable request at email ID: namalik288@gmail.com (N.A.).

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
