# Peer review of "Immunoinformatic Execution and Design of an Anti-Epstein–Barr Virus Vaccine with Multiple Epitopes Triggering Innate and Adaptive Immune Responses"

_microorganisms, 2023, doi:10.3390/microorganisms11102448_

Round 1

Reviewer 1 Report

This is a very interesting work based on predictive approaches, which integrate many bioinformatics tools and offer a very good example of work to be done for designing a vaccine. However, the manuscript suffers of a number of points to be improved.
The predicted 3D model of the constructed protein, being object of molecular simulations as MD and docking, must be evaluated carefully for the stereochemical and energetic quality, and evaluations should be presented under supplementary materials. The Ramachandran plot, although not enough to assess quality as a whole, indicates many amino acids in not allowed regions. Molecular simulations would lack significance if the initial model is of poor quality.
There is a generic use of the concept of "homology". The authors should be aware of the meaning and the discussion in literature about the wrong usage of the term (see articles with PMID codes: 3621342, 19181528). As an example, since the candidate vaccine is a construct sequence, there is no possibility that it, as a whole, shares an ancestral origin in common with any organism sequence. Any check for "similarity" is aimed to prevent side effects, but there is no reason for referring to "homology" instead of "similarity".
Add discussion of results on population coverage: as an example, how can be explained the difference in population coverage for Malaysian and World population ? Is the value for Malaysian sufficient ? Is it possible that other population have a lower coverage ?
Table 3: "The H indicates ..." I do not find "H" and it is not clear to what it is referred, and how it might be evaluated.
Fig. 6 In D-L panels, labels overlap the C-alpha and are mostly unreadable. I suggest to remove and add by graphical tools on the white background region.
Check for Typos (examples: in line 223, "b" -> "by"; in line 459  and other places "insilico" -> "in silico"; in many places, physiochemical -> physicochemical)

Minor editing is required to correct typos

Author Response

Review 1

Comments and Suggestions for Authors

This is a very interesting work based on predictive approaches, which integrate many bioinformatics tools and offer a very good example of work to be done for designing a vaccine. However, the manuscript suffers of a number of points to be improved.
The predicted 3D model of the constructed protein, being object of molecular simulations as MD and docking, must be evaluated carefully for the stereochemical and energetic quality, and evaluations should be presented under supplementary materials.

Response: Dear reviewer, we would like to appreciate your valuable comments and suggestions to our manuscript. And we would like to say thank you to you for your kind words about our study. In the current study, we have addressed all possible computational approaches which were necessary to predict the vaccine candidate. Furthermore, we have revised the manuscript according to the comments from you and other reviewers, which made the manuscript more understandable to the reader. Apart from these corrections, we have thoroughly revised the manuscript for English proofreading and grammatical mistakes.

The Ramachandran plot, although not enough to assess quality as a whole, indicates many amino acids in not allowed regions.

Response: Dear reviewer, thank you for your valuable comment on the Ramachandran plot. However, we have mention in the manuscript that the most of the amino acids (88.281%) were in the highly preferred regions, 7.812% were plotted in the preferred regions, and only 3.906% were plotted in the questionable regions. and since these are fewer than 5% AAs are plotted in the outlier section, the plot is acceptable.

Molecular simulations would lack significance if the initial model is of poor quality.

Response: Dear reviewer, we agree with your comment that the molecular simulations will be compromised if the initial model is of poor quality. However, in our study, table 2 proves that our initial model was of good quality.

There is a generic use of the concept of "homology". The authors should be aware of the meaning and the discussion in literature about the wrong usage of the term (see articles with PMID codes: 3621342, 19181528). As an example, since the candidate vaccine is a construct sequence, there is no possibility that it, as a whole, shares an ancestral origin in common with any organism sequence. Any check for "similarity" is aimed to prevent side effects, but there is no reason for referring to "homology" instead of "similarity".

Response: Dear reviewer, thank you for highlighting the error in using wrong terminology. The terminology has been corrected in the revised version of manuscript. E.g., line 161, 310, table 2.

Add discussion of results on population coverage: as an example, how can be explained the difference in population coverage for Malaysian and World population ? Is the value for Malaysian sufficient ? Is it possible that other population have a lower coverage ?

Response: Line 505-514: The discussion part has been amended as suggested. “To check which epitope cover the Malay as well as the maximum world population, we used the population coverage analysis tool integrated in the IEDB server. The individual MHC-I and MHC-II epitope coverage and combined coverage were predicted for the World and the Malay population. This was because the distribution of MHC alleles varies between geographical or ethnic groups throughout the world. Because there are over one thousand different human MHC alleles, vaccination is only effective in those who have a certain MHC allele that binds the epitope. As a result, the tool of IEDB population coverage predicted that the vaccination would cover 70.59% Malaysian population and 93.98% World population. The analysis confirmed that the vaccine candidate would be productive in all parts of the world.”

Table 3: "The H indicates ..." I do not find "H" and it is not clear to what it is referred, and how it might be evaluated.

Response: Dear reviewer, thank you for highlighting the valuable point. This was a typing error as we had both figure and table together under one heading. Actually, the table legend was mistakenly representing the part of figure. We have removed it from the table legend.

Fig. 6 In D-L panels, labels overlap the C-alpha and are mostly unreadable. I suggest to remove and add by graphical tools on the white background region.

Response: Dear reviewer, thank you for your valuable suggestion. We have revised figure 6 suing Adobe Illustrator and we believe that the figure is more understandable now.

Check for Typos (examples: in line 223, "b" -> "by"; in line 459 and other places "insilico" -> "in silico"; in many places, physiochemical -> physicochemical)

Response: Line 223, 457, 523, 524. Dear reviewer, thank you once again for highlighting the typo errors. We have revised the manuscript for grammatical mistakes and counter-checked it with the English checking software as well.

Comments on the Quality of English Language

Minor editing is required to correct typos

Response: Dear reviewer, thank you for your valuable suggestion to revise the manuscript for English proofreading and letting us know about the grammatical mistakes. The manuscript has been thoroughly revised for English proofreading and grammatical mistakes. Furthermore, to ensure the manuscript for grammatical mistakes, it has been counter-checked with the English checking software as well.

Reviewer 2 Report

The first line of the abstract is unclear. Grammar in the first line of the Introduction needs changing. English editing is required throughout and detracts from an otherwise fascinating study. Have you considered if noncoding RNAs such as miRNAs are part of the overall immune response. As you will be aware all infections and vaccines stimulate the production of varied miRNAs. This could vary with age.

Prof Christine Stabell Benn at the University of Southern Denmark has been researching the nonspecific effects of vaccines for many years. Are you able to make any comments as to how the methods used in this paper could shed light upon such nonspecific effects.

You need to provide a review of how the methods used in the study have been employed in the development of other vaccines or instances of where the methods have been employed to evaluate existing vaccines.

Poor English expression throughout.

Author Response

Review 2

Comments and Suggestions for Authors

The first line of the abstract is unclear. Grammar in the first line of the Introduction needs changing. English editing is required throughout and detracts from an otherwise fascinating study.

Response: Dear reviewer, we would like to appreciate your valuable comments and suggestions to our manuscript. We have revised the manuscript according to the comments from you and other reviewers, which made the manuscript more understandable to the reader. Furthermore, we have thoroughly revised the manuscript for English proofreading and grammatical mistakes.

Have you considered if noncoding RNAs such as miRNAs are part of the overall immune response. As you will be aware all infections and vaccines stimulate the production of varied miRNAs. This could vary with age.

Response: Dear reviewer, we did not consider if noncoding RNAs such as miRNAs are part of the overall immune response, as we didn’t find any previous computational based studies on this approach. However, this can be done in the in-vitro wet lab-based studies. We are working on this to get some possible funding so we can continue this work in our lab to prove the efficiency of this predicted vaccine based on the wet lab-based experiments including the animal studies.

Prof Christine Stabell Benn at the University of Southern Denmark has been researching the nonspecific effects of vaccines for many years. Are you able to make any comments as to how the methods used in this paper could shed light upon such nonspecific effects. You need to provide a review of how the methods used in the study have been employed in the development of other vaccines or instances of where the methods have been employed to evaluate existing vaccines.

Response: Dear reviewer, we would like to really appreciate for referring us to one of the well-known scientists in the field of vaccines. We must appreciate the efforts of Prof. Christine and her students for their efforts towards the efficacy of vaccines. Line 515-525: In the discussion section, new information has been added and the non-specific effects has been discussed as well.

Comments on the Quality of English Language

Poor English expression throughout.

Response: Dear reviewer, thank you for your valuable suggestion to revise the manuscript for English proofreading and letting us know about the grammatical mistakes. The manuscript has been thoroughly revised for English proofreading and grammatical mistakes. Furthermore, to ensure the manuscript for grammatical mistakes, it has been counter-checked with the English checking software as well.

Round 2

Reviewer 2 Report

Well done on this excellent study and best wishes for future funding. The manuscript is now suited for publication.

Author Response

Comments and Suggestions for Authors

Well done on this excellent study and best wishes for future funding. The manuscript is now suited forpublication.

Response: Dear reviewer,  we would like to appreciateonce again for your kind efforts to review the revised version of our manuscript.  And thank you for your kind words. We hope that we can contribute more significantly in the future.